# Prevalence of Fibromyalgia and Chronic Fatigue Syndrome among Individuals with Irritable Bowel Syndrome: An Analysis of United States National Inpatient Sample Database

**DOI:** 10.3390/biomedicines11102594

**Published:** 2023-09-22

**Authors:** Zahid Ijaz Tarar, Umer Farooq, Ahmad Nawaz, Mustafa Gandhi, Yezaz A. Ghouri, Asmeen Bhatt, Brooks D. Cash

**Affiliations:** 1Division of Gastroenterology and Hepatology, University of Missouri School of Medicine, Columbia, MO 65212, USA; 2Division of Gastroenterology and Hepatology, Saint Louis University, St. Louis, MO 63103, USA; umer7513781@gmail.com; 3Division of Gastroenterology and Hepatology, Suny Upstate Medical University, Syracuse, NY 13210, USA; 4Department of Medicine, University of Missouri School of Medicine, Columbia, MO 65212, USA; 5Center for Interventional Gastroenterology at UTHealth (iGUT), Division of Elective Surgery, Department of Surgery, University of Texas Health Medical School, Houston, TX 77030, USA; 6Division of Gastroenterology & Hepatology, University of Texas Health-McGovern Medical School and UT Health Science Center at Houston, St. Houston, TX 77054, USA

**Keywords:** fibromyalgia, chronic fatigue syndrome, IBS, irritable bowel syndrome

## Abstract

**Background and Aim:** Irritable bowel syndrome (IBS) is a functional gastrointestinal disorder associated with other somatic disorders. We studied the prevalence and predictors of fibromyalgia and chronic fatigue syndrome (CFS) in IBS patients. **Methods:** We used the National Inpatient Sample and included hospitalization of individuals with IBS, using ICD-10 codes, from 2016–2019. The prevalence and predictors of fibromyalgia and CFS in IBS patients were studied. Univariate and multivariate patient- and hospital-level regression models were used to calculate the adjusted odds of fibromyalgia and CFS in the IBS patient population. **Results:** Of 1,256,325 patients with an ICD-10 code of IBS included in the study, 10.73% (134,890) also had ICD-10 codes for fibromyalgia and 0.42% (5220) for CFS. The prevalence of fibromyalgia and CFS was significantly higher in IBS patients (adjusted odds ratio (AOR) 5.33, 95% confidence interval (CI) 5.24–5.41, *p* < 0.001, and AOR 5.40, 95% CI 5.04–5.78, *p* < 0.001, respectively) compared to the general adult population without IBS. IBS-diarrhea, IBS-constipation, and IBS-mixed types were independently associated with increased odds of fibromyalgia and CFS. Increasing age (AOR 1.02, 95% CI 1.01–1.04, *p* 0.003; AOR 1.02, 95% CI 1.01–1.03, *p* 0.001), female gender (AOR 11.2, 95% CI 11.1–11.4, *p* < 0.001; AOR 1.86, 95% CI 1.78–1.93, *p* < 0.001) and white race (AOR 2.04, 95% CI 1.95–2.12, *p* < 0.001; AOR 1.69, 95% CI 1.34–2.13, *p* < 0.001) were independent predictors of increased odds of fibromyalgia and CFS, respectively. **Conclusions:** It appears that IBS is associated with an increased prevalence of somatic disorders such as fibromyalgia and CFS.

## 1. Introduction

Irritable bowel syndrome (IBS) is a chronic functional disorder of the gastrointestinal tract. It is a disorder of gut–brain interaction that manifests as recurrent episodes of abdominal pain, with altered bowel habits in the absence of another obvious causative organic disease [1]. It is estimated that the worldwide prevalence of IBS is 4 to 5% [2], and that it accounts for almost 25% to 50% of referrals to gastroenterologists [3]. It is more common in women when compared to men [4] and its prevalence decreases with age [5]. While IBS does not increase overall risk of mortality [6], it has a significant effect on quality of life. Rome criteria are the most widely used criteria for the diagnosis of IBS. The most current version of these criteria, the Rome IV criteria, defines IBS as recurrent abdominal pain at least one day per week in the last three months associated with two or more of the following features: improvement in abdominal pain or discomfort with defecation, onset associated with a change in frequency of stool, and/or an onset of symptoms accompanied by a change in the form or appearance of stool [7].

Fibromyalgia is a chronic musculoskeletal pain disorder associated with fatigue, cognitive disturbances, psychiatric disorders, and multiple somatic symptoms [8,9]. In a meta-analysis by Heidari et al. [10], it was noted that 12.9% of patients with fibromyalgia have IBS as compared to 1.78% in the general population. It is more prevalent in females. The coexistence of IBS and fibromyalgia together has been linked with a greater impact on the quality of life as compared to either condition alone [11]. This is associated with more frequent fibromyalgia flares [12], and increased severity of symptoms of IBS [13]. A shared pathophysiologic mechanism has also been suggested for both IBS and fibromyalgia [14]. Chronic fatigue syndrome (CFS), also known as myalgic encephalomyelitis, is a complex multisystem disease that affects around two million Americans [15]. It is characterized by severe fatigue, cognitive dysfunction, sleep disturbance, autonomic dysfunction, and post-exertional malaise. Fatigue is also the most common extraintestinal manifestation in IBS patients. There are multiple studies that suggest a post-infectious association between IBS, CFS, and fibromyalgia [16,17,18]. The high incidence of these illnesses after infection, including GI infections, and after the use of antibiotics, has led to the hypothesis that gut dysbiosis and altered gut permeability may play a role in their pathogenesis [19,20,21].

To understand the shared burden of these somatic disorders, we queried the prevalence of fibromyalgia and CFS in patients with IBS and compared it with the general adult population. We also evaluated the predictors for having a co-diagnosis of fibromyalgia and CFS in the IBS population. To the best of our knowledge, this is the first published study utilizing a large U.S. national general population database to evaluate the burden of fibromyalgia and CFS among individuals diagnosed with IBS.

## 2. Materials and Methods

### 2.1. Study Design and Data Source

This is a retrospective cohort study of hospitalizations for the years 2016 to 2019. The selected population was based on a primary or secondary admission diagnosis of IBS with a co-diagnosis of fibromyalgia and/or CFS. The study population was selected from the national inpatient sample (NIS), a national registry of US hospital admissions, which represents more than 97% of US hospital admissions data. NIS is maintained by the agency of healthcare cost and utilization project, and it is one of the largest publicly available US healthcare databases [22]. It contains 20% stratified sample of all discharges from more than 4000 non-federal acute care hospitals in the US. In our study, we obtained records of hospital admissions from 2016–2019. Patient-level data reported in NIS are age, sex, race, household income, length of stay (LOS) and total cost of hospitalization. Hospital-level data provides information about location, bed size and teaching status of hospital. From 2016 onwards, the NIS database was coded using the International Classification of Disease, Tenth revision, Clinical Modification/Procedure Coding System (ICD-10-CM/PCS).

### 2.2. Study Population

Patients who were admitted with the principal diagnosis of IBS and a secondary diagnosis of fibromyalgia and/or CFS were included in the study. The study population was selected using the following ICD-10-CM codes: IBS with diarrhea (K58.0), IBS with constipation (K58.1), IBS-mixed type (K58.2), IBS other (K58.3), IBS without diarrhea (K58.9), fibromyalgia (M79.7) and chronic fatigue syndrome (R53.82). Patients who were younger than 18 years of age were excluded (Figure 1).

### 2.3. Study Outcomes and Variables

We determined the prevalence and predictors of fibromyalgia and CFS in IBS population, in comparison to no-IBS population. Sub-group analysis was performed on IBS-diarrhea (IBS-D), IBS-constipation (IBS-C), and IBS-mixed types. Secondary outcomes were LOS and total hospital charges. Patient- (age, sex, ethnicity, race, household income, and insurance status) and hospital-level characteristics (location, bed size, and teaching status) were included as variables of interest. Overall co-morbidity burden was determined using Deyo classification of Charlson’s co-morbidity index.

### 2.4. Statistical Analysis

We used STATA, version 18 (College Station, TX, USA) to conduct the statistical analysis. The design of the NIS database includes stratification, clustering, and weighting. Proportions were calculated using Fischer’s exact test. Univariate regression analysis was performed to calculate the unadjusted odds and multivariate regression analysis was performed to determine the adjusted odds of encountering a co-diagnosis of fibromyalgia or CFS in IBS patients. The results were adjusted for patient demographics, hospital characteristics and co-morbidities including hypertension, diabetes mellitus (DM), dyslipidemia, coronary artery disease (CAD), congestive heart failure (CHF) chronic kidney disease (CKD), cerebrovascular accident, smoking, malnutrition, alcohol abuse, and malignancy. These variables were used to construct a multivariable regression model based on significant association seen on univariate analysis with a cut-off value < 0.2 [23].

## 3. Results

We included 1,256,325 hospital admissions with a primary or secondary diagnosis of IBS in the study; out of them, 134,890 (10.73%) had fibromyalgia and 5220 (0.42%) had CFS. The IBS patients with fibromyalgia and/or CFS were younger when compared to those without fibromyalgia (58.7 vs. 62) and CFS (59.9 vs. 61.6). Majority of patients with fibromyalgia (96.5%) and CFS (89.9%) were female and white (86.5%). Similarly, CFS prevalence was highest in white population (90.7%). The prevalence of fibromyalgia varied in different income groups. It was highest in the 26th–50th percentile, when compared to 0–25th, 51st–75th and 76th–100th percentile (28.9% vs. 26.5%, 25.8%, 18.7%, respectively). The prevalence of CFS was high in 51st–75th and 76th–100th percentile (26.6%, 26.4%) when compared to 23.7% in 0–25th and 23.3% in 26th–50th percentile of median household income groups. A higher percentage of IBS patients with fibromyalgia and CFS were admitted in the Southern Hospital region (36.3%, 35.3%) followed by Midwestern (31.3%, 25.6%), Western (16.2%, 23.6%) and Northeastern (16.2%, 15.5%) regions. Most of the hospitalizations among IBS patients with fibromyalgia and CFS were at urban teaching hospitals.

Majority of the IBS patients with fibromyalgia were insured with Medicare (57.3%), followed by private/self-insurance (25.4%) and Medicaid (13.3%). Similar insurance trends were seen in patients with CFS. Hypertension was documented in 45.6% of fibromyalgia and 42.6% of CFS patients. Obesity was seen in 25.7% of fibromyalgia and 21.1% of CFS patients. Smoking was present in 24% and 23.1% of fibromyalgia and CFS patients, respectively. Other co-morbidities in fibromyalgia patients were dyslipidemia (36.2%), DM (25.1%), CAD (13.8%), CKD (11.3%), CHF (9.7%) and alcohol use (2.6%). CFS patients had a co-diagnosis of dyslipidemia (36.1%), DM (19.9%), CAD (13%), CKD (8.9%), CHF (8.7%) and alcohol use (3.4%). The baseline characteristics of the study population have been summarized in Table 1.

### 3.1. Prevalence and Adjusted Odds of Fibromyalgia and Chronic Fatigue Syndrome in Patients with Irritable Bowel Syndrome

Patients with IBS had higher prevalence of fibromyalgia in comparison to the population without IBS (10.73% vs. 1.41% *p* < 0.001). Similarly, IBS patients had a higher prevalence of CFS compared to the population without the diagnosis (0.42% vs. 0.06%, *p* < 0.001). The IBS population had increased odds of having fibromyalgia compared to the general adult population without IBS (adjusted odds ratio [AOR] 5.33, 95% confidence interval (CI) 5.24–5.41, *p* < 0.001). Similarly, IBS patients had significantly higher odds of having CFS when compared to the population without IBS (AOR 5.40, 95% CI 5.04–5.78, *p* < 0.001) (Table 2).

We performed a sub-group analysis of prevalence and odds of having fibromyalgia and CFS in patients with IBS-D, IBS-C and IBS-mixed. We concluded that IBS-D, IBS-C, and IBS-mixed populations had higher adjusted odds of having fibromyalgia and CFS independently. The adjusted odds of fibromyalgia in IBS-D, IBS-C and IBS-mixed were, respectively, AOR 4.38, 95% CI 4.20–4.57, *p* < 0.001; AOR 4.96, 95% CI 4.66–5.28, *p* < 0.001; and AOR 4.96, 95% CI 4.41–5.58, *p* < 0.001. Similarly, higher adjusted odds of CFS were seen in IBS-D, IBS-C, and IBS-mixed type (AOR 4.41, 95% CI 3.60–5.40, *p* < 0.001), (AOR 6.76, 95% CI 5.28–8.65, *p* < 0.001) and (AOR 5.32, 95% CI 3.17–8.95, *p* < 0.001), respectively (Table 3).

### 3.2. Resource Utilization

We calculated two markers of resource utilization: length of stay and total hospital charges. The LOS for fibromyalgia patients was shorter when compared to the IBS patient population without associated fibromyalgia (adjusted mean LOS −0.13, 95% CI −0.72 to 0.44, *p* = 0.64) but total charges were high by $156 (95% CI $563 to −$904, *p* = 0.68), but these results were not statistically significant. Patients with CFS showed decreased mean adjusted LOS by 0.26 days (95% CI −0.51 to −0.02, *p* = 0.06), but the results did not achieve statistical significance. Mean total hospital charges were less in CFS patients (−$5118, 95% CI −$8195 to −$2041, *p* < 0.001) (Table 4).

### 3.3. Predictors of Chronic Fatigue Syndrome and Fibromyalgia

We investigated the independent predictors of CFS and fibromyalgia. We conducted a multivariate regression analysis and concluded that with increasing age odds of fibromyalgia and CFS increases in patients with IBS (AOR 1.02, 95% CI 1.01–1.04, *p* < 0.001 and AOR 1.02, 95% CI 1.01–1.03, *p* < 0.001, respectively). Females had higher odds of fibromyalgia (AOR 11.2, 95% CI 11.1–11.4, *p* < 0.001) and CFS (1.86, 95% CI 1.78–1.93, *p* < 0.001). White race, lower socioeconomic status, smoking, alcohol use, obesity and hyperlipidemia were associated with increased odds of fibromyalgia in IBS patients, whereas black, Hispanic, and Asian race, higher socioeconomic status, and malnutrition were associated with lower odds. White race, higher socioeconomic status, smoking, obesity, and hyperlipidemia were associated with increased odds of CFS in the IBS population, whereas black, Asian, and Hispanic race, lower socioeconomic status and alcohol use were associated with lower odds of having CFS (Table 5).

## 4. Discussion

In our study, we calculated the prevalence of fibromyalgia and CFS in patients with IBS using the U.S. NIS database. In addition, we determined the predictors of having the diagnosis of these somatic diseases in IBS patients. We investigated the prevalence of fibromyalgia and CFS in IBS and the general adult population without IBS. We found that IBS patients have higher odds of having concomitant fibromyalgia and CFS when compared with the general adult population. These results agree with former studies showing that IBS patients are more prone to other somatic disorders. In a previous population-based study performed on a Danish population, prevalence rates of IBS, fibromyalgia, and CFS were 3.6%, 4.6% and 8.6%, respectively [24]. Another population study showed rates to be around 9.7%, 3.0%, and 1.3% for IBS, fibromyalgia and CFS, respectively [25]. In this analysis, we found that the prevalence rate of fibromyalgia in IBS patients was 10.7% compared to 1.4% in the general adult population. In the scientific literature, the range of prevalence rates of fibromyalgia in IBS patients varies from 12.9% to 77% [10,12,26,27,28], which is higher than the general population. The wide range of prevalence is most likely because of heterogenous study designs, differences in the medical acuity level of study centers, incomparable populations, and lack of uniform diagnostic criteria for both IBS and fibromyalgia. Our study showed a higher rate of CFS in IBS patients when compared to the general adult population. Similar findings were shown in a study by Whitehead et al. [26].

In our study, we found out that IBS patients with fibromyalgia as well as IBS patients with CFS were younger compared to IBS patients without these conditions. It is an interesting observation, as having multiple somatic comorbidities at a younger age can be very debilitating. We discovered that increasing age was found to be associated with increased odds of having fibromyalgia and CFS in IBS patients. Female sex has been an established risk factor for IBS, as well as for fibromyalgia and CFS [4,5,29,30,31,32,33]. This coincides with our study which also showed that being a female with IBS was associated with having fibromyalgia and CFS. Our study also showed that obesity was associated with increased odds of having fibromyalgia and CFS in the IBS population. Several mechanisms including altered small bowel and colonic transit, gut microbiota changes, low fiber intake, and high refined carbohydrate diet have been linked to obesity and associated with IBS symptoms [34,35], but no clear association has been established between IBS and obesity. However, some common pathogenic pathways have been associated with obesity, CFS and fibromyalgia, including hypothalamic-pituitary-adrenal axis dysregulation and excessive activation of the sympathetic nervous system [36,37,38,39]. Smoking was another modifiable risk factor in our study that was associated with increased chances of having fibromyalgia and CFS in the IBS population. It has been reported in literature that fibromyalgia patients who smoke tend to report more pain and worse overall health compared to non-smokers [40,41].

A study by Woolley et al. [42] on CFS patients observed that almost two-thirds of patients reduce or stop alcohol intake altogether after being diagnosed with CFS, and the most common reason was increased fatigue after drinking. In our study, we observed that alcohol use was associated with decreased odds of having CFS in IBS patients. This may be because CFS patients did not consume alcohol as it exacerbates fatigue, rather than alcohol having any protective effect against CFS. We observed that in IBS patients, lower socioeconomic status was associated with increased odds of having fibromyalgia, whereas higher socioeconomic status was associated with increased odds of having CFS. A similar trend has been noted regarding fibromyalgia and CFS in the general population by Collin et al. [43]. A possible explanation of this dissimilar pattern observed in fibromyalgia when compared to CFS could be due to selection bias (CFS patients often requiring specialist service for diagnosis—an obstacle for patients with lower socioeconomic status) as well as due to a lower prevalence of CFS in ethnic minorities [44].

Many patients with somatic disorders have extreme deconditioning [45,46,47] due to lack of exercise, that can lead to increased low-density lipoprotein (LDL) levels. This increase in LDL was demonstrated by Gurer et al. in fibromyalgia patients [48]. Our study also showed that hyperlipidemia is associated with increased odds of developing fibromyalgia and CFS in the IBS population. We observed that Caucasians have a significantly higher incidence of fibromyalgia and CFS compared to other races. Most of the IBS patients with fibromyalgia and CFS were insured via Medicare. The LOS and total hospital charges for fibromyalgia patients were not statistically different when compared to the control group, and similarly, there was no statistical difference in LOS for CFS patients, but it was noted that mean total hospital charges were less for CFS patients.

Based on predominant bowel habits, there are four subtypes of IBS: IBS-C, IBS-D, IBS with mixed bowel habits, and IBS unclassified (that cannot be accurately categorized into one of the other three subtypes). We performed sub-group analysis with IBS-D, IBS-C and IBS-mixed type, and concluded that all subtypes were independently associated with having increased odds of fibromyalgia and CFS when compared to the general adult population. The odds of having IBS-C or mixed type were higher when compared to IBS-D. These results are echoed by Erdrich et al., who showed that predominantly mixed and constipation types of IBS were associated with fibromyalgia [49].

It has been reported that fibromyalgia patients with IBS have worse symptoms of pain, fatigue, and morning tiredness as compared to those without IBS [11], and during the flare of fibromyalgia they have aggravated gastrointestinal symptoms [11,12,13,14,15,16,17,18,19,20,21,22,23,24,25,26,27,28,29,30,31,32,33,34,35,36,37,38,39,40,41,42,43,44,45,46,47,48,49,50]. The severity of the IBS symptoms has been shown to have a positive association with fibromyalgia. Patients with a mild form of IBS frequently did not carry a diagnosis of fibromyalgia, whereas patients with moderate-to-severe IBS symptoms had an increased association with fibromyalgia [12,13,14,15,16,17,18,19,20,21,22,23,24,25,26,27,28]. This observation raises the possibility that alterations in the gut microbiome may play a role in pain symptoms. Research is currently underway exploring the gut–brain axis as a driver of unexplained pain syndromes [51,52]. Sensitization of the central nervous system and sympathetic nervous system predominance is another proposed mechanism that is linked with different somatic conditions like fibromyalgia, CFS and IBS [53,54,55]. Earlier identification of comorbidities in IBS patients will be valuable, since they can be managed with better treatment strategies. In current clinical practice, there is a high risk of neglecting multi-syndromic patients. We as clinicians should integrate in our practice with regular screening for other somatic disorders in the IBS population and determine the need to consult other specialties like rheumatology and psychiatry to improve the overall health outcome in IBS patients. A study by Basnayake et al. showed that a multidisciplinary care approach is superior to the gastroenterologist-only approach in managing functional gastrointestinal disorders [56].

The NIS is a large database which increases power of the study but it has certain limitations. First of all, its retrospective design can determine an association of disease but not a causal relationship. Furthermore, lack of blinding and randomization creates bias. It does not provide medication and laboratory data; so, we lack the ability to measure the effect of pharmacological therapies. In addition, few of the confounding factor are missed due to unavailability of the ICD-10 codes for them.

In conclusion, IBS patients have higher prevalence of somatic comorbidities like fibromyalgia and chronic fatigue syndrome. Both IBS-C and IBS-D were independently associated with increased odds of having fibromyalgia and chronic fatigue syndrome; odds were higher among IBS-C patients when compared to those with IBS-D. Increasing age, female sex, white race, obesity, smoking, and hyperlipidemia were predictors for having a co-diagnosis of both fibromyalgia and CFS in IBS patients. The identification and treatment of these disorders can improve their quality of life.

## Figures and Tables

**Figure 1 biomedicines-11-02594-f001:**
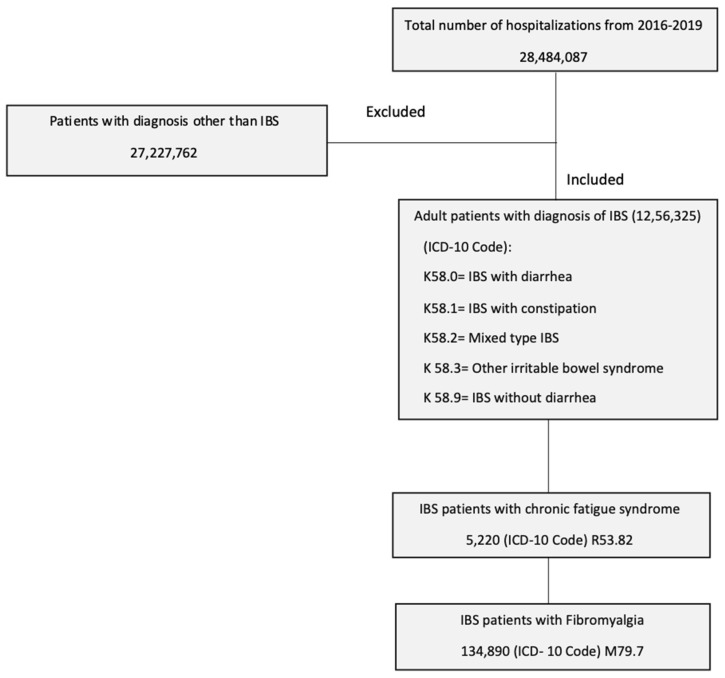
Patient selection process using the national inpatient sample.

**Table 1 biomedicines-11-02594-t001:** Baseline characteristics of the study population.

Baseline Characteristics	Total Number of IBS Patients 1,256,325 (1.04%)		*p*-Value Fibromyalgia	*p*-Value Chronic Fatigue Syndrome
	Fibromyalgia *n* = 134,890 (10.73%)	Chronic Fatigue syndrome *n* = 5220 (0.42%)		
Mean Age [years]	58.7 (58.5–59.0)	59.9 (58.9–60.8)	<0.001	0.0003
Women [*n* (%)]	130,169 (96.5%)	(89.9%)	<0.001	<0.001
Race [*n* (%)]			<0.001	<0.001
White	116,680 (86.5%)	4735 (90.7%)		
Black	9200 (6.82%)	198 (3.8%)		
Hispanic	6111 (4.53%)	181 (3.46%)		
Asians	391 (0.29%)	IS		
Native Americans	540 (0.41%)	26 (0.49%)		
Others	1915 (1.42%)	72 (1.38%)		
Charlson Co-Morbidity Index [*n* (%)]			<0.001	<0.001
0	33,722 (25.0%)	1639 (31.4%)		
1	39,118 (29.0%)	1389 (26.6%)		
2	25,494 (18.9%)	867 (16.6%)		
3 or more	36,555 (27.1%)	1326 (25.4%)		
Median Household Income in Zip code (Quartile) *			<0.001	<0.001
1st (0–25th)	35,746 (26.5%)	1237 (23.7%)		
2nd (26th–50th)	38,983 (28.9%)	1216 (23.3%)		
3rd (51st–75th)	34,802 (25.8%)	1389 (26.6%)		
4th (76th–100th)	25,224 (18.7%)	1378 (26.4%)		
Hospital Region [*n* (%)]			<0.001	<0.001
Northeast	21,852 (16.2%)	809 (15.5%)		
Midwest	42,221 (31.3%)	1336 (25.6%)		
South	48,965 (36.3%)	1843 (35.3%)		
West	21,852 (16.2%)	1232 (23.6%)		
Insurance Status [*n* (%)]			<0.001	<0.001
Medicare	77,292(57.3%)	3001 (57.5%)		
Medicaid	17,940(13.3%)	441 (8.45%)		
Private/Self pay	34,262 (25.4%)	1561 (29.9%)		
Uninsured	2104 (1.56%)	81 (1.54%)		
Hospital bed size [*n* (%)]			0.21	0.0001
Small	29,001 (21.5%)	1138 (21.8%)		
Medium	37,095 (27.5%)	1430 (27.4%)		
Large	69,199 (51.3%)	2652 (50.8%)		
Hospital teaching status [*n* (%)]			0.007	0.009
Rural	12,410 (9.2%)	449 (8.6%)		
Urban non-teaching	29,811 (22.1%)	1211 (23.2%)		
Urban teaching	92,669 (68.7%)	3560 (68.2%)		
Co-Morbidities				
Smoking	32,374 (24.0%)	1206 (23.1%)	0.001	0.20
Alcohol abuse	3507 (2.6%)	178 (3.4%)	<0.001	0.81
Obesity	34,667 (25.7%)	1101 (21.1%)	<0.001	0.60
Hypertension	61,240 (45.4%)	2234 (42.6%)	<0.001	0.96
Dyslipidemia	48,830 (36.2%)	1884 (36.1%)	0.003	0.53
Diabetes Mellitus	33,857 (25.1%%)	1039 (19.9%)	<0.001	0.009
Congestive heart Failure	13,084 (9.7%)	454 (8.7%)	<0.001	0.01
Coronary Artery Disease	18,615 (13.8%)	679 (13.0%)	<0.001	0.003
Cerebro-Vascular Accident	324 (0.24%)	35 (0.67%)	0.01	0.05
Chronic kidney Disease	15,243 (11.3%)	465 (8.9%)	<0.001	<0.001
Malnutrition	5800 (4.3%)	266 (5.1%)	<0.001	0.50
Malignancy	4856 (3.6%)	256 (4.9%)	<0.001	0.95

* First quartile: $1–$42,999, $1–$43,999, $1–$45,999 and $1–$47,999 for NIS 2016, 2017, 2018 and 2019 respectively; 2nd quartile: $43,000–$53,999, $44,000–$55,999, $46,000–$58,999 and $48,000–$60,999 for NIS 2016, 2017, 2018 and 2019 respectively; 3rd quartile: $54,000–$70,999, $56,000–$73,999, $59,000–$78,999 and $61,000–$81,999 for NIS 2016, 2017, 2018 and 2019 respectively; 4th quartile: > $71,000, >$74,000, >$79,000 and >$82,000 for NIS 2016, 2017, 2018 and 2019, respectively.

**Table 2 biomedicines-11-02594-t002:** Adjusted odds of fibromyalgia and chronic fatigue syndrome in patients with irritable bowel syndrome.

Outcomes	Adjusted Odds Ratio	95% CI	*p*-Value
**Fibromyalgia**	5.33	5.24–5.41	<0.001
**Chronic fatigue syndrome**	5.40	5.04–5.78	<0.001

**Table 3 biomedicines-11-02594-t003:** Adjusted odds of fibromyalgia and chronic fatigue syndrome in patients with IBS-D and IBS-C.

Outcomes	IBS-Diarrhea		IBS-Constipation		IBS-Mixed	
	AOR	*p*-Value	AOR	*p*-Value	AOR	*p*-Value
**Fibromyalgia**	4.38 (4.20–4.57)	<0.001	4.96 (4.66–5.28)	<0.001	4.96 (4.41–5.58)	<0.001
**Chronic fatigue syndrome**	4.41 (3.60–5.40)	<0.001	6.76 (5.28–8.65)	<0.001	5.32 (3.17–8.95)	<0.001

IBS-D, Irritable Bowel Syndrome Diarrhea type; IBS-C, Irritable Bowel Syndrome Constipation type; AOR, Adjusted Odds Ratio.

**Table 4 biomedicines-11-02594-t004:** Resource utilization in admitted patients with irritable bowel syndrome.

Outcomes	Fibromyalgia		CFS	
	AOR (95%CI)	*p*-Value	AOR (95%CI)	*p*-Value
**Adjusted mean LOS**	−0.13 (−0.72 to 0.44)	0.64	−0.26 (−0.51 to −0.02)	0.06
**Adjusted mean total hospital charges**	156$ (−563 $ to −904$)	0.68	−5118$ (−8195$ to −2041$)	0.001

**Table 5 biomedicines-11-02594-t005:** Predictors of chronic fatigue syndrome and fibromyalgia in IBS population.

Predictors		CFS		Fibromyalgia		
	AOR	95% CI	*p* Value	AOR	95% CI	*p* Value
**Age**	1.02	1.01–1.03	<0.001	1.02	1.01–1.04	<0.001
**Female sex**	1.86	1.78–1.93	<0.001	11.2	11.1–11.4	<0.001
**White race ***	1.69	1.34–2.13	<0.001	2.04	1.95–2.12	<0.001
**Black race**	0.75	0.59–0.97	0.003	0.47	0.46–0.49	0.02
**Hispanic**	0.60	0.47–0.76	<0.001	1.03	0.98–1.08	0.17
**Asian**	0.61	0.45–0.82	<0.001	0.31	0.29–0.34	<0.001
**Low socioeconomic status * (1st–25th) quartile**	0.81	0.76–0.87	<0.001	1.35	1.32–1.38	<0.001
**26th–50th quartile**	1.11	1.04–1.17	<0.001	0.97	0.96–0.99	<0.001
**51st–75th quartile**	1.22	1.15–1.29	<0.001	0.89	0.87–0.91	<0.001
**76th–100th quartile**	1.22	1.14–1.30	<0.001	0.73	0.72–0.75	<0.001
**Smoking**	1.14	1.09–1.19	<0.001	1.24	1.22–1.25	<0.001
**Obesity**	1.30	1.24–1.36	<0.001	1.86	1.84–1.88	<0.001
**Malnutrition**	1.09	0.99–1.01	0.08	0.86	0.84–0.88	<0.001
**Alcohol use**	0.80	0.73–0.88	<0.001	1.09	1.06–1.12	<0.001
**Hyperlipidemia**	**1.28**	**1.23–1.33**	**<0.001**	**1.30**	**1.29–1.32**	**<0.001**

CFS: Chronic fatigue syndrome; AOR: Adjusted Odds Ratio. * As a reference.

## Data Availability

On individual request, data can be provided after approval from Agency of healthcare Cost and Utilization Project.

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
