# Peer review of "Prevalence of Fibromyalgia and Chronic Fatigue Syndrome among Individuals with Irritable Bowel Syndrome: An Analysis of United States National Inpatient Sample Database"

_biomedicines, 2023, doi:10.3390/biomedicines11102594_

Round 1

Reviewer 1 Report

The presented work concerns hospitalized patients with a primary admission diagnosis of irritable bowel syndrome(IBS). This confirms the reports of many other authors about the co-occurence of IBS and fibromyalgia(FM) and chronic fatigue syndrome(CFS). However, it do not confirms causal relationship between them. Especially, that the patients suffered from other serious diseases, as diabetes, hypetension, heart failure and other.  The large number of IBS patients qualified for hospitalization also raises doubsts, because they are usually treated on an outpatient basis.

Among predictors of IBS and FM and CFS obesity and malnutrition were included without taking into account eating habits. The proper diet and dysbiosis plays a very important role in the pathogenesis of the above disease syndrome, especially all type of chronic fatigue (peripheral, infection and central fatigue).

I suggest knowledge of this topic to be supplemented in the discussion section before publication.

Author Response

Point 1: The presented work concerns hospitalized patients with a primary admission diagnosis of irritable bowel syndrome(IBS). This confirms the reports of many other authors about the co-occurence of IBS and fibromyalgia(FM) and chronic fatigue syndrome(CFS). However, it do not confirms causal relationship between them. Especially, that the patients suffered from other serious diseases, as diabetes, hypetension, heart failure and other.  The large number of IBS patients qualified for hospitalization also raises doubsts, because they are usually treated on an outpatient basis.

Response: 

We agree with the reviewer that it does not determine a causal relationship, that's why we reported the prevalence of FM and CFS in the IBS patient population. We also agree that IBS is managed outpatient, that was the reason we selected all the patients who had an ICD-10 code for IBS, not necessarily admitted for IBS. NIS has a diagnosis code from 1-40 so we took all those patients who have a history of IBS or were admitted for IBS. We adjusted our analysis for other major and serious diseases including diabetes, HTN, heart failure, etc.

Comment 2: 

Among predictors of IBS and FM and CFS obesity and malnutrition were included without taking into account eating habits. The proper diet and dysbiosis plays a very important role in the pathogenesis of the above disease syndrome, especially all type of chronic fatigue (peripheral, infection and central fatigue).

Response:

This is a limitation of NIS it does not provide data on eating habits, and we mentioned that in the revised manuscript.

Reviewer 2 Report

The study on the relationship between IBS and fibromyalgia  / CFS may have some interest once it is well specified, starting from the title, which is not a population study but a study on hospitalized patients, therefore selected on the basis of clinical criterion, probably the severity of the symptoms. Since the vast majority of IBS patients are not hospitalized in their lifetime (at most they access PS service) this needs to be well discussed by the authors.

I suggest changing the title to: “Prevalence and predictors of Fibromyalgia and Chronic Fatigue Syndrome among patients hospitalized for Irritable Bowel Syndrome in US" (or something similar).

The discussion must be adequate by eliminating any comparison with population studies, specifying the context well. The same must happen in the conclusions and abstract.

A comparison of the data could make sense compared to other populations with functional syndromes in the course of hospitalization (eg: headache).

Author Response

Comment 1:The study on the relationship between IBS and fibromyalgia  / CFS may have some interest once it is well specified, starting from the title, which is not a population study but a study on hospitalized patients, therefore selected on the basis of clinical criterion, probably the severity of the symptoms. Since the vast majority of IBS patients are not hospitalized in their lifetime (at most they access PS service) this needs to be well discussed by the authors.

Response: We agree with the reviewer that IBS I mostly managed on an Outpatient basis; that is the reason we titled our project as a population-based study because we have included all the patients who had a diagnosis of IBS, not admitted for IBS. Even though the patient was hospitalized for other reasons and had a history of IBS, we analyzed how many of these patients also had CFS and FM.

Comment 2: 

The discussion must be adequate by eliminating any comparison with population studies specifying the context well. The same must happen in the conclusions and abstract.

Response: As we described above, we conducted this study as population analysis, and this was the reason to compare it with other population studies. We made the relevant changes on a few points in the discussion in the revised manuscript. We changed the hospitalization with IBS to patient admitted for any reason but also had an ICD-0 code for IBS.

Comment 3:The discussion must be adequate by eliminating any comparison with population studies, specifying the context well. The same must happen in the conclusions and abstract.

Response: We had made the relevant changes in the discussion.

Round 2

Reviewer 2 Report

I welcome the corrections but I think that the title should make it clear that this is a population of hospitalized patients and not a general or outpatient population.

Author Response

Comment: I welcome the corrections but I think that the title should make it clear that this is a population of hospitalized patients and not a general or outpatient population.

Reply: We have adjusted the title for clarity. Thank you.